# The Extrapolar SWIFT model (version 1.0): Fast stratospheric ozone chemistry for global climate models

Daniel Kreyling[1], Ingo Wohltmann[1], Ralph Lehmann[1], and Markus Rex[1]

[1]Alfred-Wegener-Institute Helmholtz-Centre for Polar and Marine Research

*Correspondence to:* Daniel Kreyling (daniel.kreyling@awi.de)

**Abstract.** The Extrapolar SWIFT model is a fast ozone chemistry scheme for interactive calculation of the extrapolar strato-spheric ozone layer in coupled general circulation models (GCMs). In contrast to the widely used prescribed ozone, the SWIFT ozone layer interacts with the model dynamics and can respond to atmospheric variability or climatological trends.

The Extrapolar SWIFT model employs a repro-modelling approach, where algebraic functions are used to approximate the numerical output of a full stratospheric chemistry and transport model (ATLAS). The full model solves a coupled chemical differential equations system with 55 initial and boundary conditions (mixing ratio of various chemical species and atmospheric parameters). Hence the rate of change of ozone over 24 h is a function of 55 variables. Using covariances between these variables, we can find linear combinations in order to reduce the parameter space to the following nine *basic* variables: latitude, pressure altitude, temperature, overhead ozone column, mixing ratio of ozone and of the ozone depleting families ($Cl_y$, $Br_y$, $NO_y$ and $HO_y$). We will show that these 9 variables are sufficient to characterize the rate of change of ozone. An automated procedure fits a polynomial function of fourth degree to the rate of change of ozone obtained from several simulations with the ATLAS model. One polynomial function is determined per month which yields the rate of change of ozone over 24 h. A key aspect for the robustness of the Extrapolar SWIFT model is to include a wide range of stratospheric variability in the numerical output of the ATLAS model, also covering atmospheric states that will occur in a future climate (e.g. temperature and meridional circulation changes or reduction of stratospheric chlorine loading).

For validation purposes, the Extrapolar SWIFT model has been integrated into the ATLAS model replacing the full strato-spheric chemistry scheme. Simulations with SWIFT in ATLAS have proven that the systematic error is small and does not accumulate during the course of a simulation. In the context of a 10 year simulation, the ozone layer, simulated by SWIFT, shows a stable annual cycle, with inter-annual variations comparable to the ATLAS model. The application of Extrapolar SWIFT requires the evaluation of polynomial functions with 30 – 100 terms. Nowadays, computers can calculate such polynomial functions at thousands of model grid points in seconds. SWIFT provides the desired numerical efficiency and computes the ozone layer $10^4$ times faster than the chemistry scheme in the ATLAS CTM.

## 1 Introduction

Modern climate models include an increasing number of climate processes and run with ever higher model resolutions. Many processes that are relevant for the climate system, are already well understood, but they remain computationally too demanding

to be incorporated into climate models. One of these processes is the stratospheric ozone chemistry. The feedbacks between the ozone layer and the changing climate system have been investigated in various studies (e.g., Thompson and Solomon, 2002; Rex et al., 2006; Baldwin et al., 2007; Nowack et al., 2014; Calvo et al., 2015). All of them emphasize the importance of the interactions between climate change and the ozone layer. Climate simulations with a more accurate representation of the ozone

layer lead to significant changes in tropospheric and surface variables. However, a frequently used approach to represent the ozone layer in general circulation models (GCMs) is the use of prescribed zonal mean ozone climatologies, as in many of the Coupled Model Intercomparison Project 5 (CMIP5) simulations (IPCC, 2014). By using prescribed ozone, the atmospheric dynamics cannot interact with the ozone field, the ozone hole is a static, zonally symmetric feature, that does not interact with atmospheric waves and the ozone layer does not respond to climate change and vice versa. But this approach is computationally

cheap and does not impede the GCM's capacity for ensemble simulations. The incorporation of an interactive ozone layer instead of climatologies allows the ozone field to actually match the model dynamics and enables two-directional feedbacks. Chemistry Climate Models (CCMs) with a highly resolved stratosphere usually provide such an interactive ozone layer, but the computational cost of CCMs still limits their usefulness for long-term ensemble simulations (Eyring et al., 2010). In recent years different approaches were taken to efficiently incorporate interactive ozone in climate simulations (Eyring et al., 2013).

One of these approaches is the development of stratospheric ozone chemistry schemes with a very low computational burden in comparison to the computation time of the GCM, e.g. the Cariolle scheme (Cariolle and Teyssedre, 2007) or the Linoz scheme (Hsu and Prather, 2009). In this paper we introduce the extrapolar part of the numerically efficient and interactive stratospheric ozone chemistry scheme SWIFT. Its goal is to provide sufficient accuracy and efficiency to enable ensemble simulations with atmosphere-ocean coupled GCMs, while maintaining the physical and chemical properties of the processes that govern ozone

chemistry in the stratosphere, so that the SWIFT approach is valid for a wide range of climatic conditions, including future climate scenarios.

SWIFT is subdivided into a polar and an extrapolar module. The two sub-modules follow separate approaches due to the differences in polar and extrapolar ozone chemistry. The lack of sunlight and very low temperatures during polar night extend the chemical lifetimes of various trace gases, relevant for ozone depletion. Under these conditions the individual species within

the chemical families $Cl_y, Br_y, NO_y$ and $HO_y$ are too far from chemical equilibrium, so that their time evolution needs to be calculated with differential equations. The Polar SWIFT model simulates the time evolution of polar-vortex averaged mixing ratios of ozone and 4 key species during Arctic and Antarctic winters. A small coupled differential equation system, containing empirically determined fit parameters, models the most relevant processes of polar ozone depletion. The first Polar SWIFT version was described by Rex et al. (2014) and the updated version was published by Wohltmann et al. (2017).

In extrapolar conditions the diurnal average concentrations of the individual species within the chemical families (partitioning) mentioned above are sufficiently close to photochemical steady state because the photochemical lifetimes of the involved species are sufficiently short compared to the transport timescales. In a good approximation the chemically induced change in ozone over 24 hours is a function of the concentrations of the chemical families, ozone itself and the physical boundary conditions. The Extrapolar SWIFT model is based on the substitution of a comprehensive differential equation system describing the

ozone changes by algebraic functions. This approach is also referred to as repro-modelling and has been successfully applied

to chemical models, e.g. Spivakovsky et al. (1990); Turányi (1994); Lowe and Tomlin (2000). As in the previous studies we obtain the algebraic functions by fitting the numerical solution of the chemical differential equation system with orthonormal polynomial functions. Following the approach of Turányi (1994) we use a wide range of input and output values of a full chemical model to create a data set that is then used for fitting the polynomial functions. However, a few modifications were introduced, most prominently, in the selection of the most suitable polynomial terms. Moreover, we developed a termination criterion that does not require the selection of arbitrary thresholds. It is important to note that the repro-model is not a shortened subset of the full chemical system. By approximating the output of the full system, we ensure that all physical and chemical properties of the full chemical model are maintained in the repro-model. In this application the rate of change of ozone in the lower and middle stratosphere is parameterized by one polynomial function per month. Each of these polynomials is a function of 9 *basic* variables, which are sufficient to parameterize the rate of change of ozone in the full chemical system. The *basic* variables are latitude, pressure altitude, temperature, the overhead ozone column, the volume mixing ratio (VMR) of the ozone depleting substances (ODS) combined into 4 chemical families and ozone itself. The calculation of the polynomial function values instead of solving the chemical differential equation system drastically reduces the computational cost and makes SWIFT a suitable candidate for coupling to a GCM.

Existing fast ozone schemes for climate models like the Cariolle scheme (Cariolle and Teyssedre, 2007) or the Linoz scheme (McLinden et al., 2000; Hsu and Prather, 2009) use a first order Taylor-series expansion of the rate of change of ozone around mean atmospheric states of ozone mixing ratio, temperature and the overhead ozone column. In comparison to SWIFT, these schemes do not explicitly include the abundance of ODS as a variable in the model. Handling changes in stratospheric ODS abundance requires the repeated determination of production and loss rates and their derivatives. Including the ODS as additional degrees of freedom in the Extrapolar SWIFT model, increases its resilience towards ODS variability. Moreover, the linear Taylor-series functions tend to produce larger deviations where the rate of change of ozone is not linear with respect to the variability of the three variables. The Extrapolar SWIFT polynomial functions are continuous throughout the stratosphere and can cope with the non linear parts of the rate of change of ozone.

In section 2 of this paper the application of repro-modelling to the rate of change of ozone is described. First we introduce the set-up of the repro-model, containing polynomial coefficients as free parameters. Further the approximation algorithm determining these coefficients is described and its modifications in comparison to previous studies are explained. Section 3 focuses on the domain of definition of the polynomial functions and how outliers are handled in Extrapolar SWIFT. A validation and error estimation of the polynomial functions are presented in section 4. Eventually, two different simulations with SWIFT are discussed in section 5. A 2-year simulation focuses on the error in the ozone field caused by the monthly polynomial functions. A 10-year simulation mimics the set-up of SWIFT in a GCM and demonstrates the stability of the model over a longer simulation period. The development of Extrapolar SWIFT and the results of the simulations are also discussed in Kreyling (2016).

## 2 Application of repro-modelling to stratospheric ozone chemistry

### 2.1 Setting up the repro-model

The Extrapolar SWIFT repro-model calculates the rate of change of ozone over 24 h by evaluating polynomial functions of fourth degree. Each polynomial function is valid during one month of the year. To determine these polynomial functions we
use multivariate fitting of a representative data set which comprises a wide range of stratospheric conditions, as suggested by Turányi (1994). As a source for the rate of change of ozone we use the comprehensive Lagrangian stratospheric chemistry and transport model ATLAS. The ATLAS model is described in detail in Wohltmann and Rex (2009) and Wohltmann et al. (2010). It contains 49 stratospheric trace gases interacting with each other in over 170 gas phase and heterogeneous chemical reactions. Together with atmospheric and geographic initial and boundary conditions the differential equation system contains
55 variables and parameters. The rate of change of ozone may be represented as a function of 55 arguments:

$$\frac{dO_x}{dt} = \hat{F}(x_1, x_2, \ldots, x_{55}) \tag{1}$$

where $O_x$ is the VMR of the odd-oxygen family, containing $O_3, O$ and $O(^1D)$ and $\hat{F} : \mathbb{R}^{55} \to \mathbb{R}$. The $O_x$-family has a longer chemical lifetime than ozone which is beneficial to our approximation approach. Moreover, in the lower and middle stratosphere odd-oxygen almost entirely consists of ozone. Thus in Extrapolar SWIFT $O_x$ substitutes $O_3$.

In order to set up a repro-model, we need to determine a set of *basic* variables which are sufficient for the parameterization of all the physical and chemical processes in the full chemical system. The determination of *basic* variables is a crucial aspect since their number should be large enough, so that the function in Eq. 1 is approximated with sufficient accuracy. On the other hand, their number should be as small as possible, so that the repro-model is numerically efficient. This is partly achieved by lumping the chemical species into chemical families. The following 4 chemical families are relevant for ozone depletion in the
stratosphere and therefore constitute 4 of the *basic* variables:

$$Cl_y = \underbrace{Cl + ClO + Cl_2O_2}_{short-lived} + \underbrace{ClONO_2 + HCl}_{reservoir}$$

$$Br_y = \underbrace{Br + BrO + HBr + HOBr}_{short-lived} + \underbrace{BrONO_2 + BrCl}_{reservoir}$$

$$NO_y = \underbrace{N + NO + NO_2 + NO_3}_{short-lived} + \underbrace{HNO_3}_{reservoir}$$

$$HO_y = \underbrace{H + OH + HO_2}_{short-lived} + \underbrace{H_2O}_{reservoir}$$

The stratospheric ozone depletion is driven by catalytic cycles involving the short-lived species of the above-listed chemical families. Consequently, the repro-model requires information on the concentration of the short-lived compounds. These may

be derived from the concentrations of the chemical families. In the extrapolar regions the short-lived reactive species (e.g. $ClO_x$ or $BrO_x$) are sufficiently close to chemical equilibrium determined by the local conditions (e.g. pressure, temperature, radiation and the abundance of reaction partners). Consequently, in the chemical families containing only one reservoir gas ($NO_y$ and $HO_y$) the concentration of the short-lived species is uniquely determined by the abundance of the total family, i.e. we assume local chemical equilbrium between the short-lived and reservoir species. For $Cl_y$ and $Br_y$ the partitioning between the reservoir species needs to be considered. However, in most regions of the extrapolar stratosphere the life-time of $ClONO_2$ is shorter than the time scales of vertical or meridional transport, so that $ClONO_2$ also comes close to equilibrium state. The same can certainly be assumed for $BrONO_2$, which has an even shorter life-time than $ClONO_2$.

Apart from the VMR of the chemical constituents, the reaction rates depend on temperature, air density and in the case of photolysis rates on the actinic flux and particularly on the ultra-violet-attenuation (UV-attenuation). These parameters must also be implicitly or explicitly included into the set of *basic* variables. Table 1 summarizes the 9 *basic* variables we have identified. The column 'Remarks' points out different properties and processes parameterized by the variable. A function of these 9 variables (Eq. 2) sufficiently approximates the function in Eq. 1, but reduces the dimensionality from 55 to 9.

$$\Delta O_x = F(\phi, z, T, topO_3, Cl_y, Br_y, NO_y, HO_y, O_x) \tag{2}$$

where $\Delta O_x$ is the rate of change of ozone over 24 h and $F : \mathbb{R}^9 \to \mathbb{R}$. After determining an approximation for $F \approx \tilde{F}$ in Eq. 2 by SWIFT, the chemical change of the ozone VMR at each grid point in a GCM simulation can be calculated by Eq. 3:

$$O_x(t+24h) = O_x(t) + \Delta O_x(t) * 24h$$
$$\approx O_x(t) + \tilde{F}(\phi(t), z(t), T(t), topO_3(t), Cl_y(t), Br_y(t), NO_y(t), HO_y(t), O_x(t)) * 24h \tag{3}$$

## 2.2 Approximation algorithm

The algebraic equation of the repro-model is a polynomial function of fourth degree (i.e. the sum of the exponents of a term is $<= 4$). The polynomial uses the same 9 *basic* variables as in Eq. 2 and yields the rate of change of ozone over 24 h. The $\Delta O_x$-function in Eq. 2 can be approximated by a polynomial function $\tilde{F}$:

$$F(x_1, \ldots, x_9) \approx \tilde{F}(x_1, \ldots, x_9) = \sum_{j=1}^{n} c_j f_j(x_1 \ldots, x_9), \tag{4}$$

where $x_1, \ldots, x_9$ are the *basic* variables, $f_j$ are polynomial terms (e.g. $f_1 = x_1^2 x_2$) and $c_j$ are their coefficients for $j = 1 \ldots n$ where $n$ is the number of polynomial terms. For a polynomial function of fourth degree with 9 variables, the maximum number of terms is 715, including all mixed-terms. The coefficients $c_j$ in Eq. 4 are determined such that the rate of change of $O_x$ is calculated by the ATLAS CTM for $m$ different values of the *basic* variables $x_{i1}, \ldots, x_{i9}$, $i = 1, \ldots, m$, are approximated with best accuracy:

$$F(x_{i1}, \ldots, x_{i9}) \approx \sum_{j=1}^{n} c_j f_j(x_{i1}, \ldots, x_{i9}) \tag{5}$$

**Table 1.** 9 *basic* variables of the Extrapolar SWIFT model. The column "Remarks" lists properties and processes parameterized by the variable. Pressure altitude is defined as $z = -H \log(\frac{p}{p_0})$ and overhead ozone is the integrated ozone column above a specific location in the atmosphere.

| Variable | Unit | Symbol | Remarks |
|---|---|---|---|
| Latitude | [°] | $\phi$ | solar zenith angle and actinic flux |
| Pressure altitude | [m] | z | air density and actinic flux |
| Temperature | [K] | T | kinetics of reactions |
| Overhead ozone column | [DU] | $topO_3$ | attenuation of UV-radiation |
| Chlorine family | [ppb] | $Cl_y$ | catalytic $ClO_x$-chemistry |
| Bromine family | [ppt] | $Br_y$ | catalytic $BrO_x$-chemistry |
| Nitrogen oxide family | [ppb] | $NO_y$ | catalytic $NO_x$-chemistry |
| Hydrogen oxide family | [ppm] | $HO_y$ | catalytic $HO_x$-chemistry |
| Odd-oxygen | [ppm] | $O_x$ | Chapman and catalytic chemistry |

The $m$ different values of the *basic* variables will be referred to as training data points or training data set. In order to write Eq. 5 in matrix notation we define a $m \times n$ matrix $\mathbf{A}$ with $\mathbf{A}_{ij} = f_j(x_{i1}, \ldots, x_{i9})$ and a vector $\mathbf{F}$ with $F_i = F(x_{i1}, \ldots, x_{i9})$, $i = 1, \ldots, m$. The polynomial coefficients $c_j$ are grouped into a vector $\mathbf{c}$. Then, the linear equation system in Eq. 5 can be expressed as

$$\mathbf{Ac} = \mathbf{F} \quad . \tag{6}$$

To determine $\mathbf{c}$ we employ the least squares method, which is to minimize the Euclidian-norm ($\|\,\|$) of the deviation between the approximation and $\mathbf{F}$:

$$\|\mathbf{Ac} - \mathbf{F}\| \rightarrow \min \tag{7}$$

The minimization in Equation 7 can be made more efficient and numerically stable by first transforming the matrix $\mathbf{A}$ into an orthogonal matrix. Spivakovsky et al. (1990) achieve this with successive Householder transformations which finally yield the $\mathbf{QR}$-decomposition of matrix $\mathbf{A}$. Turányi (1994) and Lowe and Tomlin (2000) use the Gram-Schmidt process for orthogonalization. The literature suggests (e.g. Golub and Van Loan, 1996), that the unmodified Gram-Schmidt process has worse numerical properties which can impair the orthogonalization. In our approach we are using a $\mathbf{QR}$-decomposition based on the Householder transformation.

We start the fitting procedure with one polynomial term ($n = 1$) on the right hand side of Eq. 5. During the following iterations the polynomial function is consecutively extended by one additional term. This corresponds to an extension of the matrix $\mathbf{A}$ by one column. Turányi (1994) started the approximation with the constant term and continued with linear terms, then quadratic terms and so on up to terms of maximum degree, also including all mixed terms. In each iteration the residuum

$\|\mathbf{Ac} - \mathbf{F}\|$ was calculated. If the current residuum was reduced by a certain threshold relative to the previous residuum, then the current term got accepted to be added to the polynomial function. This method tested the terms in a given arbitrary order. If the order of testing had been a different one, other polynomial terms would be accepted and the overall quality of the polynomial function could potentially be better.

5 In our approach we are circumventing this problem by testing all polynomial terms individually as next additional term. I.e., in each iteration each of the still available polynomial terms is temporarily added to the already selected terms and the fitting procedure is carried out. The term which reduces the residuum the most is permanently added to the polynomial function and removed from the pool of available terms. In the next iteration all remaining terms are fitted in combination with the previously accepted ones. By simply choosing the best fitting term we also avoid setting an arbitrary threshold for the minimum required

10 reduction of the residuum. This polynomial term selection method makes the fitting procedure computationally much more extensive. However, the fitting procedure has to be carried out only once, so that this additional computation time imposes no disadvantage during the application of SWIFT.

 The more polynomial terms are added to the function, the better the approximation will be, i.e. the residuum can be reduced further and further. If as many polynomial terms (corresponding to columns of $\mathbf{A}$) are fitted as there are training data points

15 (corresponding to rows of $\mathbf{A}$) then the linear equation system in Eq. 6 is not overdetermined anymore. In this case any small scale structure, originating from the random distribution of training data points would have been fitted and the polynomial function would contain an impractically large number of terms. An over-fitted polynomial function actually causes the residuum to be higher, when it is applied to an independent data set (i.e. not a subset of the data the polynomial was fitted to). Consequently the fitting procedure should be terminated before the random fluctuations in the training data set get fitted. This termination

20 criterion can be defined by applying the selected polynomial terms and their coefficients to an independent data set instead of the training data set. The independent data set is named testing data set here. The quality of the approximation is expressed by

$$r = \|\mathbf{A}^{\mathrm{Test}}\mathbf{c} - \mathbf{F}^{\mathrm{Test}}\|, \tag{8}$$

where $\mathbf{A}^{\mathrm{Test}}$ is like the matrix $\mathbf{A}$, only the rows of $\mathbf{A}^{\mathrm{Test}}$ correspond to the testing data points and the vector $\mathbf{F}^{\mathrm{Test}}$ contains the rate of change of ozone at the testing data points. The polynomial coefficients $c$ are the ones determined via Eq. 7. $r$ is

25 the residuum corresponding to the polynomial function with one temporarily added term. At some iteration, during the fitting procedure, the residuum $r$ will not be reduced by any of the available additional terms. This defines the termination of the approximation algorithm.

 It is important that the testing data set has the same probability distribution of *basic* variables as the training data set. We achieve this by randomly separating the output of the ATLAS simulations into the training and the testing data set, containing

30 2/3 and 1/3 of the total output, respectively.

## 2.3 Latitude and altitude boundaries of the repro-model

In this section we discuss where in the stratosphere the Extrapolar SWIFT model can be used, i.e. for which latitudes and altitudes the underlying assumptions are valid. A key aspect for the definition of this latitude-altitude region is the mean

chemical lifetime $\tau$ of $O_x$.

$$\tau = \frac{[O_x]}{R} \tag{9}$$

where $[O_x]$ is the concentration instead of VMR and $R$ is the sum of the rates of all $O_x$ depleting catalytic cycles. In Figure 1 the mean chemical lifetime of $O_x$ taken from ATLAS data for January is displayed. The contour labels specify the lifetime in days.

In the lower stratosphere the $O_x$ lifetimes exceed 365 days. The longer lifetimes in the lower stratosphere are a consequence of the slower reaction rates of the catalytic $O_x$-loss cycles mostly due to less O atoms. The O atom is produced via the photolysis of $O_3$, but its vertical distribution is mainly controlled by the three-body reaction $O + O_2 + M \rightarrow O_3 + M$. The rate constant of this reaction increases with increasing pressure. The latitudinal (and seasonal) variation of the $O_x$ lifetime reflects the varying length of the day and the attenuation of solar radiation on its way through the atmosphere.

Above roughly 30 km altitude the mean lifetime of $O_x$ is shorter than vertical and meridional transport time scales. In this quasi chemical equilibrium state the $O_x$ concentration is determined by the local meteorological conditions and the abundance of the ODS. Consequently $O_x$ can be calculated as a function $F_{eq}$ of the previously mentioned *basic* variables, but without the $O_x$ VMR itself, so that $F_{eq} : \mathbb{R}^8 \rightarrow \mathbb{R}$.

$$O_x = F_{eq}(\phi, z, T, topO3, Cl_y, Br_y, NO_y, HO_y) \tag{10}$$

Accordingly, in the upper stratosphere the $O_x$ VMR at a point in time $t + \Delta t$ is a function of 8 *basic* variables at time $t + \Delta t$. The function $F_{eq}$ can also be approximated by a polynomial function $\tilde{F}_{eq}$:

$$O_x(t + \Delta t) \approx \tilde{F}_{eq}(x_1(t + \Delta t), \ldots, x_8(t + \Delta t)), \tag{11}$$

where $x_1, \ldots, x_8$ correspond to the 8 variables in equation 10. In SWIFT the polynomial functions calculate $\Delta O_x$ and determine the $O_x$ VMR of the next time step as in Eq. 3. The $O_x$ VMR at time $t + \Delta t$ is a function of the 9 *basic* variables $x_1(t), \ldots, x_9(t)$.

$$O_x(t + \Delta t) = O_x(t) + \tilde{F}(x_1(t), \ldots, x_9(t)) * 24h \tag{12}$$

Both equations (Eq. 11 and Eq. 12) yield $O_x(t + \Delta t)$. Setting them equal results in:

$$O_x(t) + \tilde{F}(x_1(t), \ldots, x_9(t)) * 24h = \tilde{F}_{eq}(x_1(t + \Delta t), \ldots, x_8(t + \Delta t))$$
$$\tilde{F}(x_1(t), \ldots, x_9(t)) * 24h = \tilde{F}_{eq}(x_1(t + \Delta t), \ldots, x_8(t + \Delta t)) - O_x(t) \tag{13}$$

This means that in the quasi equilibrium region of $O_x$, $\tilde{F}$ is the result of the $O_x$ polynomial function in equilibrium $\tilde{F}_{eq}$ minus the linear $O_x$ term. However, the polynomial function $\tilde{F}$ contains various $O_x$ terms of higher degree. These terms together with the higher $O_x$ VMR in the upper stratosphere causes rather large errors. Consequently, the polynomial function $\tilde{F}$ is not suited to be used in the region where $O_x$ is in quasi chemical equilibrium. The altitude of 30 km roughly marks the transition between the equilibrium and non-equilibrium state of $O_x$. The lifetime is roughly 14 days in this altitude (see Figure 1). We

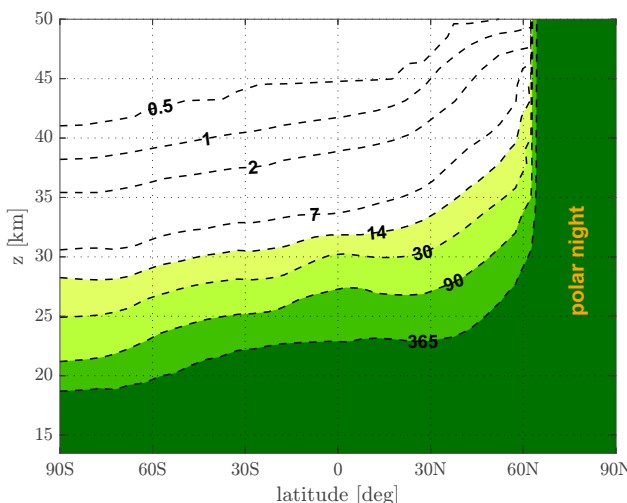

**Figure 1.** Zonal mean of $O_x$ lifetime in January, derived from ATLAS CTM data. Contour numbering shows the lifetime in days.

defined the 14 days contour to be the upper boundary, up to where the polynomial functions can be used or rather up to where the training and testing data sets reach.

Since the lifetime of $O_x$ is a function of the incoming solar radiation, the altitude and tilt of the 14 days contour also depends on the season. In the course of the year the tilt of the lifetime contours will shift. For each monthly polynomial function

we defined a separate upper boundary. In the quasi equilibrium region (upper stratosphere) the SWIFT simulations currently require ozone values, interpolated from stratospheric climatologies. For the future a similar repro-modelling approach will be applied to function $F_{eq}$ in Eq. 10, by fitting the $O_x$ VMR directly instead of the rate of change of $O_x$.

However, the upper stratosphere only contributes a few percent to the stratospheric ozone column. The bulk of ozone, dominating the total column values, is in the lower stratosphere, below 30 km. This motivated our focus on this part of the

stratosphere which we will refer to as $\Delta O_x$-regime.

The lower boundary of the $\Delta O_x$-regime is set to 15 km pressure altitude (roughly 120 hPa). In the tropics 15 km is approximately the altitude of the tropical tropopause layer (TTL) defining the boundary between tropospheric and stratospheric air. In the extratropical regions ozone rich air can also be found below 15 km, especially in the northern high latitudes. However, at theses altitudes and latitudes the rate of change of ozone is close to zero and the transport of ozone is much more relevant (see

also Figure 1). When running Extrapolar SWIFT in a GCM, treating ozone as a passive tracer below 15 km pressure altitude is recommended.

The regime boundaries between Extrapolar SWIFT and Polar SWIFT are defined by the edge of the polar vortex. The horizontal extent of the polar vortex is defined by 36 mPV units, where mPV is the modified potential vorticity according to Lait (1994) (with $\theta_0 = 475K$). In the vertical, the specified vertical extent of the Polar SWIFT domain goes from roughly

18 km to 27 km pressure altitude. Above and below Polar SWIFT the extrapolar module is used, although the rate of change of ozone is close to zero during polar night.

## 2.4 Training data

The monthly training and testing data set for Extrapolar SWIFT are generated with the stratospheric Lagrangian chemistry and transport model ATLAS (Wohltmann and Rex, 2009; Wohltmann et al., 2010). The data used in this work originated from two 2.5 years simulations, one from November 1998 to March 2001 and the second from November 2004 to March 2007. The chemistry module of ATLAS contains a comprehensive set of gas phase chemical reactions and a heterogeneous chemistry scheme. Photolysis and reaction coefficients are taken from the recent Jet Propulsion Laboratory (JPL) catalog (Sander et al., 2011). All partial species of the 4 ozone depleting chemical families ($Cl_y$, $Br_y$, $NO_y$ and $HO_y$) are included in the 49 ATLAS species. The individual species are initialized from different sources. The VMR of $H_2O$, $N_2O$, HCl, $O_3$, CO and $HNO_3$ were initialized from Aura Microwave Limb Sounder (MLS) climatologies for the 1998–2001 simulation. The 2004–2007 simulation already used the measurements of Aura MLS directly (Waters et al., 2006). The VMR of $CH_4$ and $NO_2$ (substitute for $NO_x$) were taken from climatologies of the HALogen Occultation Experiment (HALOE) instrument (Grooß and Russell III, 2005). Initial values for $Cl_y$ and $Br_y$ were derived from tracer-tracer-correlations to $CH_4$ and $N_2O$ measured during an aircraft and ballooning campaign described in Grooß et al. (2002). The ATLAS trajectories are initialized in roughly 2 km thick pressure altitude layers with an horizontal resolution of 200 km. On each trajectory the chemistry is calculated like in a chemical box model. ATLAS solves a coupled system of differential equations to obtain the rate of change of the trace gases. The stiff numerical solver uses an automatic adaptive time step and is based on the Numerical Differentiation Formulas (Shampine and Reichelt, 1997). After 24 h (mixing time step) the mixing algorithm merges or creates trajectories and interpolates the chemical species accordingly. The ATLAS trajectories are driven by ERA-Interim wind fields, temperatures and heating rates (Dee et al., 2011).

For each month of the year, daily snapshot values of the *basic* variables at the current location of the trajectories and the corresponding $\Delta O_x$ at a fixed time of day (00 UTC) are compiled into a data set which is later split into training and testing data set. The number of trajectories computed in an average ATLAS run is roughly $10^5$ throughout the lower and middle stratosphere. In order not to exceed the size of the computer's main memory, a random sub-sample of the $10^5$ trajectories of each day of a month was taken. This was done, so that all monthly data sets contain the same number of data: 8 million and 4 million data points in the training and testing data set, respectively. The monthly data sets are chosen such that they also contain a fraction of data from the 10 days preceding and following the current month. We do this in order to ensure a smoother transition of polynomial functions from one month to the next.

Individual chemical species in ATLAS are grouped into their respective families and summed up to generate the mixing ratios of $Cl_y$, $Br_y$ and $NO_y$. $HO_y$ is simply substituted by water vapor, since the $H_2O$ VMR is a factor thousand larger than the sum of all other $HO_y$ constituents. The $\Delta O_x$-value is defined by the difference of the $O_x$-VMR between two snapshots along the Lagrangian trajectory. A $\Delta O_x$-value is associated with the beginning of a 24 h period:

$$\Delta O_x(t) * 24h = O_x(t + 24\ h)) - O_x(t) \tag{14}$$

Before the fitting procedure, the *basic* variables are normalized to a range from 0 to 1. Otherwise the order of magnitude of the polynomial coefficients would vary extremely due to the strongly varying magnitude of the *basic* variables (e.g. pressure altitude $\approx 10^4$ m versus $Br_y$ VMR $\approx 10^{-11}$).

The Lagrangian trajectories in ATLAS are not distributed homogeneously. In general, higher trajectory densities can be
found where there is strong horizontal and vertical wind shear, e.g. at the edge of the polar vortex. This is caused by the trajectory mixing algorithm in ATLAS, which initializes new or deletes existing trajectories based on their rate of divergence or convergence in a region of the model atmosphere. The regions of increased trajectory densities coincide with strong gradients of chemical constituents and meteorological parameters. Thus these gradients are well resolved in ATLAS, which is beneficial to Extrapolar SWIFT. The training and testing data sets simply contain the same, unmodified, sampling as in ATLAS and
therefore also resolve the gradients well.

## 3 Validity of repro-model in a changing climate

### 3.1 Domain of definition of polynomial functions

The extensive training data set derived from the ATLAS CTM fills a portion of the 9 dimensional hyper space, which defines the domain of definition of the fitted polynomial functions. SWIFT is intended to be used in long-term climate simulations
and it will certainly encounter inter-annual and decadal variability. Therefore we used data from ATLAS simulations covering a wide range of stratospheric variability. By taking the training and testing data from different decades we include maximum and minimum conditions of the solar cycle. The data also represents different QBO-phases as well as the varying strengths and lifetimes of the Arctic and Antarctic polar vortices.

Climatological changes impacting the probability distribution of the *basic* variables can also be expected, e.g. changes in
temperature and meridional circulation. The resilience of SWIFT to such trends is outlined in Figure 2. Future climate scenarios will shift the current probability density function (PDF) of the *basic* variables. The schematic in Figure 2 shows a shift of the temperature PDF, assuming a normal distribution of the temperature, with the 8 other *basic* variables fixed. Most of the PDF in the training climate (blue) and the future climate (orange) overlaps. The slightly colder conditions of the future climate are thus mostly covered by the present domain. Only at low temperatures, where the probability is small, outliers (red) occur. These
outliers will force the polynomial functions to extrapolate and likely produce erroneous $\Delta O_x$-values. Therefore outliers need to be identified and the extrapolation must be prevented.

Apart from a PDF shift like the one illustrated in Figure 2 there can be scenarios, where the shift of the PDF is too severe and the repro-model can not be applied. An example would be the reduction of stratospheric chlorine by 50 %. The majority of the $Cl_y$-PDF would be outside the original PDF. In such a case the repro-model needs to be re-fitted to an adjusted training
climate, which can easily be done by running the full ATLAS model for a few years, driven by output from a climate model or with modified levels of the ODS.

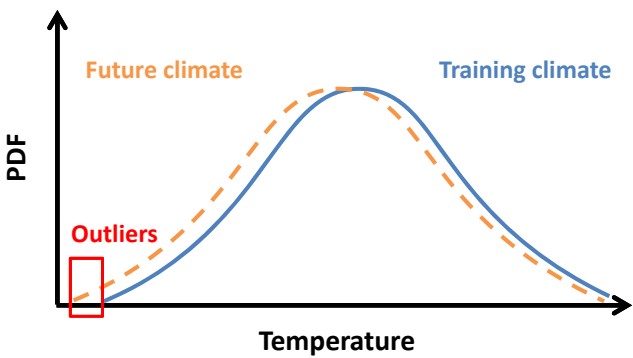

**Figure 2.** Schematic of a shift in the probability density function of stratospheric temperature in a future climate.

## 3.2 Handling outliers

When running a SWIFT simulation, the polynomial function should not be evaluated outside the domain defined by the training data set. Polynomial functions of higher degree tend to rapidly increase or decrease when extrapolated. In order to determine if a data point lies outside or inside the 9 dimensional domain of definition we need to be able to define its boundaries. This
could be achieved by enveloping the 9 dimensional cloud of data points by a conjunction of 9 dimensional cells (cuboids), corresponding to a 9 dimensional regular grid (look-up table). These grid cells are either sampled by the training data set or not. A sampled grid cell is defined as being inside the domain, all the not-sampled grid cells are outside. Dealing with a 9 dimensional grid with only a few nodes per dimension readily creates a grid with millions of cells. However, the majority of these grid cells represent combinations of *basic* variables which do not occur in the stratosphere (e.g. warm temperatures
in the lower most stratosphere). Consequently less than 0.1 % of the grid cells actually get sampled by the training data set. Using efficient ways to store and search this sparse data set would be a feasible option for identifying outliers. However, in our approach we make use of the regular grid but go one step further. Again we employ a fitting procedure to determine a polynomial function that yields positive values inside the sampled domain and negative values outside. This polynomial function is hereafter called domain-polynomial. The regular grid, sampled by the training data set will be referred to as training
grid and is used for fitting of the domain-polynomial. The domain-polynomial is obtained in the following way. First the cells of the training grid get assigned either positive or negative values. The positive values (inside the domain) are derived from the number of neighboring cells also sampled as being inside the domain. Outside the domain the cells get assigned negative values derived from the cell's distance to the closest cell inside the domain. In order to improve the quality of the fit at the domain boundary, some smoothing operations were applied. By removing individual cells being isolated in the opposing region the
transition from positive to negative values becomes more smooth. Additionally we removed outside-cells which are adjacent to one cell inside the domain, but not to any other. These cells get assigned values of only -1 but are actually surrounded by outside-cells with much lower values. Finally the grid cells which were assigned values close to zero are copied multiple times in the training grid, to increase the weight of this region during the fit.

During the application of SWIFT within a GCM the following operations are carried out at each spatial grid point. The domain-polynomial is computed for the values of the 9 *basic* variables, in order to determine whether these values reside inside or outside the domain of definition of the original polynomial function. If inside, the $\Delta O_x$ is calculated as usual. If the values of the 9 *basic* variables prove to be outside, we need to determine a close location inside the domain of definition, where a $\Delta O_x$ can be calculated safely. Newton's method is applied to find a nearby null of the domain-polynomial, which defines the boundary of the domain of definition. Within a certain margin of the null ($\pm 0.5$) the iteration of Newton's method is stopped and the $\Delta O_x$-value is calculated at the current coordinates in the 9 dimensional space. An advantage of using the domain-polynomial is that it's derivatives can be computed easily and used in Newton's method.

## 4 Validation of polynomial functions

### 4.1 Comparison of the rate of change of ozone

As an initial validation step the rate of change of ozone in the testing data set is compared to the rate of change of ozone calculated by the polynomial functions. In Figure 3 the $\Delta O_x$ in ATLAS and Extrapolar SWIFT is displayed as zonal averages. The ATLAS $\Delta O_x$ is taken from the testing data sets and the SWIFT $\Delta O_x$ from the polynomial functions evaluated on the testing data set. The 4 months shown (January, April, July and October) are selected as representative of each season. The data are binned into equivalent latitude (5°) versus pressure altitude (1 km) bins and averaged. Grey shaded bins either mask areas outside the $\Delta O_x$-regime (e.g. polar vortex, upper or lower regime boundary) or indicate too few trajectories to yield a meaningful average. Since the effective area of the zonal bands decreases towards the poles, the bins with too few trajectories are found in high latitudes.

In general all 4 months show good agreement between ATLAS and SWIFT. Especially in the tropics and mid-latitudes the amplitude of $\Delta O_x$ and the extent of regions of production or loss compare very well. Even detailed structures like the two local maxima in the tropical ozone production region in January are visible in SWIFT. Steep gradients of $\Delta O_x$, e.g. around 25 km at mid-latitudes are well reproduced by the polynomial functions. Deviations between ATLAS and SWIFT occur at the upper boundary of the summer hemisphere and in high latitudes at the beginning of the winter season (e.g. southern hemisphere in April, northern hemisphere in October). In the 9 dimensional hyper space some boundary regions of the training and testing data set are less densely populated with trajectories than more central regions. This can have different reasons, but the most obvious one is the spatial difference of the trajectory density caused by the mixing algorithm in ATLAS (see 2.4). Moreover the extreme values of some of the 9 *basic* variables occur less frequently if they are approximately Gaussian distributed. Finally the selection criteria for the trajectories described in Section 2.3 can cause sparsely sampled regions. E.g. due to the variability of the polar vortex in one year the January data set will include certain polar latitudes, which will not be included in the January of the next year. In regions with lower trajectory density fewer squared errors need to be minimized during the least-squares minimization. Consequently these regions have less weight in the approximation than more densely sampled regions and the deviations will be larger. However, we decided not to manipulate the trajectory density in the training and testing data sets,

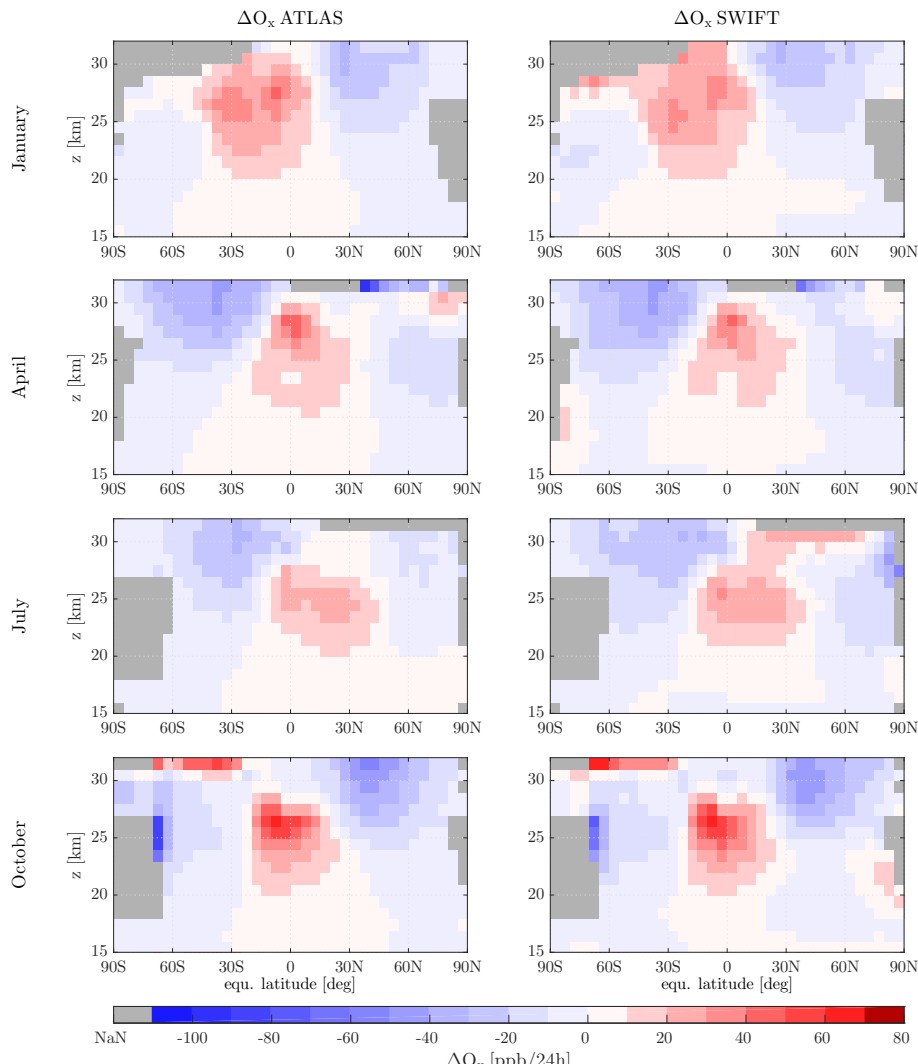

**Figure 3.** Zonal and monthly mean of $\Delta O_x$ from the testing data sets for 4 representative months. Left column ATLAS $\Delta O_x$, right column the result of the SWIFT polynomial functions evaluated at the testing data. Grey bins contain no or too few data.

because we wanted to maintain the frequency with which meteorological and chemical conditions occur in ATLAS. A sparsely populated region in the 9 dimensional space implies infrequent and therefore less relevant stratospheric conditions.

## 4.2 Error estimation

To estimate the error of Extrapolar SWIFT, we examine the difference of ATLAS $\Delta O_x$ minus SWIFT $\Delta O_x$ divided by the $O_x$

5  VMR.

$$Q = \frac{\Delta O_{xSWIFT} - \Delta O_{xATLAS}}{O_x} \times 100 \tag{15}$$

$Q$ is given in units of percent per day [%/day]. $Q$ describes the positive or negative percentage drift of $O_x$ VMR per day, due to the error in SWIFT. The division by $O_x$ makes the differences at small and high $O_x$ VMR more comparable, instead of just interpreting the absolute deviation. Similar to the relative error, $Q$ tends to have larger values for very small $O_x$ VMR. These properties of $Q$ need to be taken into account, when considering different regions with high or low ozone VMR. In the

lower tropical stratosphere where very small $O_x$ VMR can be found, the absolute errors of SWIFT are small in contrast to the $Q$-values which can exceed $\pm 50$ %/day. However, for the calculation of the total ozone column the deviations at small $O_x$ VMR are irrelevant. Also for the computation of the atmospheric heating rates based on the SWIFT ozone field, the absolute errors originating from other green house gases (e.g. $CO_2$), with a much higher concentration are much more important, than the deviations at small $O_x$ VMR. In Figure 4 we will discuss the distribution of $Q$. Later, in Section 5 we will use the absolute

deviations between the SWIFT and the ATLAS simulation to discuss the error quantitatively.

Figure 4 shows the probability distribution of $Q$ for the 4 representative months January, April, July and October. As in Figure 3 the $Q$-values of the roughly 4 million data points of each monthly testing data set are discussed. The bin width of one bar in Figure 4 is 0.2 %/day. Thus over 20 % of $Q$-values reside within the interval of $\pm 0.1$ %/day in all 4 months. The majority of $Q$-values lies within the $\pm 1$ %/day interval. The mean (pink dashed line) and the median (cyan dotted line) are close to zero.

The strongest systematic biases (mean) are $-0.3$ %/day in July and $+0.25$ %/day in October, the median however is centered very close to $0.0 \ \%/day$ in both months. The grey shaded area shows the standard deviation (STD) around the mean. The variability of the STD indicates that the quality of the approximation actually varies significantly between the months. The errors of the October polynomial function (STD of 3.5 %/day) are spread more strongly than the errors of the April polynomial function (STD is roughly 0.6 %/day).

As mentioned before, individual $Q$-values can surpass $\pm 50$ %/day where the $O_x$ VMR is small, i.e. below 100 ppb. But these extreme deviations are rare, which is demonstrated by the 5 % and 95 % quantiles (black dotted lines). 90 % of the total $Q$-values are located in between the 2 quantile-lines.

## 5   Simulations with SWIFT

### 5.1   SWIFT coupled to the ATLAS CTM

The Extrapolar SWIFT module was coupled to the ATLAS CTM, in order to perform validation simulations. In this set up the SWIFT scheme replaces the detailed stratospheric chemistry model of ATLAS. Apart from the geographical and meteorological variables provided by ATLAS, Extrapolar SWIFT requires the VMR of the 4 ozone depleting chemical families $Cl_y, Br_y, NO_y$ and $HO_y$. We compiled monthly zonal climatologies to be distributed with the model if required. The $H_2O$ climatology (substituting the $HO_y$ family) is based on extensive observational data from Aura MLS. The $Cl_y, Br_y$ and $NO_y$ climatologies

are composed of the 2 ATLAS simulations used in the training and testing data sets. All species in ATLAS contributing to one of the chemical families are summed up and weighted according to their yield of active chlorine, bromine or $NO_x$, respectively. The initialization of chemical species for the 2 ATLAS simulations was described in Section 2.4. For initialization and regions

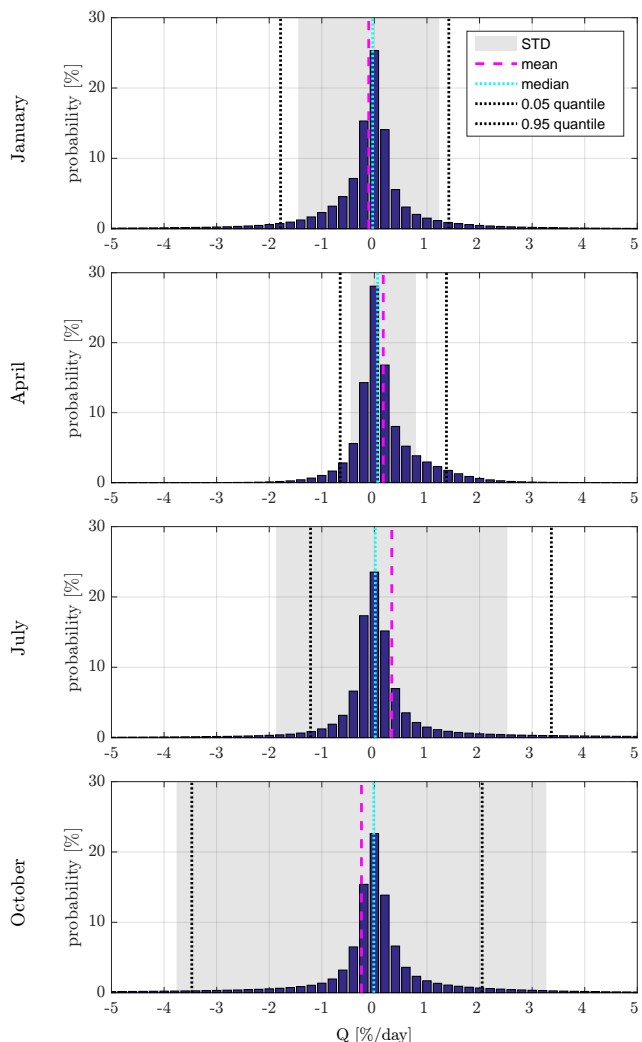

**Figure 4.** Probability distribution of error quantity $Q$ of monthly testing data sets January, April, July and October. Grey shading indicates the standard deviation around the mean (pink dashed line). The dotted lines shows the median (cyan) and 5 % and 95 % quantile (black).

outside the $\Delta O_x$-regime an additional $O_x$ climatology is required. This climatology is also compiled of an extensive set of ozone measurements by Aura MLS.

The SWIFT in ATLAS simulations are driven by ERA-Interim data (Dee et al., 2011). Every 24 h the SWIFT module is called and the rate of change of ozone is calculated based on the current conditions at the beginning of each trajectory. The VMR of the 4 ozone depleting chemical families are interpolated from the trace gas climatologies. Latitude, pressure altitude and atmospheric temperature are defined by the trajectory and the overhead ozone column is integrated from the ozone values of the overhead trajectories. In combination with these 8 parameters, the ozone VMR of the last time step (24 h before) is used to calculate the rate of change of ozone ($\Delta O_x$) by evaluating the polynomial function. Eventually the $\Delta O_x$ is added to the $O_x$

VMR from the last time step, according to Equation 3. In order to smooth the transition between two polynomial functions corresponding to consecutive months, we linearly interpolate between the $\Delta O_x$-results of the two polynomial functions. All other components of the ATLAS CTM, like the trajectory transport or the mixing algorithm remain unchanged. The SWIFT in ATLAS simulations apply outlier handling as described in Section 3.2.

Above the seasonally dependent upper boundary of the $\Delta O_x$-regime, as introduced in Section 2.3, climatology values of $O_x$ are used in the simulation. In a layer that extends over 2 km below this upper boundary the $O_x$ VMR are determined by computing an altitude-weighted average between values from the climatological $O_x$ values and the $\Delta O_x$-regime. Inside the polar vortex $O_x$ climatology values are used. The Polar SWIFT module is intentionally switched off to investigate only the performance of the extrapolar module.

## 10   5.2   2-year simulation

Initially, the Extrapolar SWIFT module coupled to ATLAS was used in a simulation over a period of 2 years. With this short simulation we want to compare the development of the ozone layer in SWIFT to a reference simulation with ATLAS. The goal of the comparison is to investigate the error or drift caused solely by the SWIFT polynomial functions. Therefore the simulation conditions of both runs should be as similar as possible. To achieve this, the SWIFT simulation does not use

trace gas climatologies for $Cl_y, Br_y, NO_y, H_2O$ and $O_x$, but uses zonally and daily averaged trace gas VMR instead. These daily values are compiled from the reference ATLAS simulation. Thus, apart from the averaging, the background trace gas fields are identical in both simulations. Further, the simulation covers a 2 year time period which coincides with the period from which half of the training data originated (years 2005 and 2006). By selecting this simulation period we ensure that the SWIFT polynomial functions were trained with the stratospheric conditions of those years. I.e. the errors can not be caused by

stratospheric variability unknown to SWIFT.

The panels in Figures 5 and 6 show monthly averaged ozone concentrations for the 2-year SWIFT simulation (middle column). The reference ATLAS simulation is shown in the left column and the difference between both in the right column. Since it is the ozone concentrations and total ozone columns that are crucial for the feedback of ozone to the model's radiation, we have transformed the mixing ratios produced by SWIFT into ozone concentrations here. In the regions outside the $\Delta O_x$-

regime, e.g. inside polar vortex (white contour) or above the upper boundary (black dashed line), $O_x$-values from the daily averaged $O_x$-fields are used.

Figure 5 shows the entire annual cycle of 2005 (first simulation year) in bi-monthly intervals. Figure 6 repeats the sequence for the second simulation year, 2006. Throughout both years SWIFT shows excellent agreement with the ozone layer of the ATLAS simulation. The seasonal cycle of the ozone layer is very well reproduced. The average deviation oscillates between

$\pm 0.2 \times 10^{12}/cm^3$. Over the course of the year 2005 the positive differences in the lower stratosphere of the northern hemisphere change sign to negative differences in the second half of the year. This pattern can also be observed in the second simulation year 2006. If the polynomial functions produce similar deviations in the same month of different years, we can attribute the deviations to a suboptimal approximation. However, the discussed deviations are in a region of strong meridional transport, where the residence time of air parcels is sufficiently short so that no significant accumulation of errors occurs.

Further, it is unlikely for the monthly polynomial functions to produce the same deviations in exactly the same regions. If we compare the magnitude of the positive differences in January and March 2005 versus January and March 2006 we see that the more positive deviations have switched from one month to the other. The variability of the magnitude can probably be attributed to the inter-annual stratospheric variability of the northern hemisphere, in particular the extent and lifetime of the polar vortex. In general the deviations of year 2006 are not larger or more extensive than in 2005. Apparently no significant error is propagated from the preceding year to the following year.

## 5.3   10-year simulation

A SWIFT simulation over a period of 10 years demonstrates the stability of the model. The set up for this simulation mimics the coupling of SWIFT to a GCM, although SWIFT is actually running in the ATLAS CTM. The trace gas climatologies for $Cl_y, Br_y, NO_y$ and $H_2O$ are the monthly climatologies described in Section 5.1. The simulation starts in November 1998 and continues until December 2008. This period encompasses both training data periods, the time between the two and a period after the last training data period. The bright blue curve in Figure 7 shows the seasonal and inter-annual variation of the stratospheric ozone layer simulated by SWIFT. The depicted value is the integrated stratospheric ozone column in Dobson units from 15 km to 32 km pressure altitude. In order to observe a strong seasonal signal, we choose to display a location in the northern hemispheric mid-latitudes (Potsdam at 52.4° North, 13.0° East). The orange and green shaded years in Figure 7 are the simulation periods of the training data set. The red curve in both periods shows the values of the reference ATLAS simulation. In both periods SWIFT reproduces the seasonal signal seen in ATLAS quite well. Especially in the green shaded patch the agreement between SWIFT and ATLAS seems to be as good as in the orange patch, although SWIFT was running continuously for 4 years in between. To demonstrate this more clearly, the scatter plot in Figure 8 shows daily averaged ozone columns of SWIFT on the X-axis versus the ones from ATLAS on the Y-axis. The coloring of the dots corresponds to the two time periods in Figure 7. The scatter of data points from both periods overlaps entirely and the magnitude and distribution of deviations from the diagonal is identical. Clearly the errors of SWIFT did not accumulate over the course of the previous 6 years.

Beginning in autumn 2004 observational data from the microwave limb-sounder Aura-MLS is available and we additionally compare the SWIFT results with the Aura-MLS observations (black line in Figure 7). In autumn 2005 and 2006 ATLAS underestimates the ozone columns in comparison to the Aura-MLS observations. Since SWIFT is trained with ATLAS data, SWIFT also reproduces this underestimation of about 30 DU and continues underestimating the autumn stratospheric ozone columns in the years 2007 and 2008 (pink shaded patch). During the first half of each year, however, SWIFT matches the Aura-MLS columns quite well and even captures the inter-annual variability shown by the observations (compare spring maximum 2007 vs. 2008). The scatter plot in Figure 9 shows daily averaged ozone columns from the green and pink shaded years. Some amount of deviation in this Figure is also caused by the difference in geo-location between the MLS-profile and the selected location in the SWIFT simulation (Potsdam). Days when no MLS measurement was taken in a 200 km radius of Potsdam are excluded, which reduces the total amount of days by about 50 %. Again the coloring of the dots corresponds to the periods in the time series (Fig. 7). As already seen in the monthly means in Fig. 7, SWIFT underestimates the smaller ozone columns (autumn

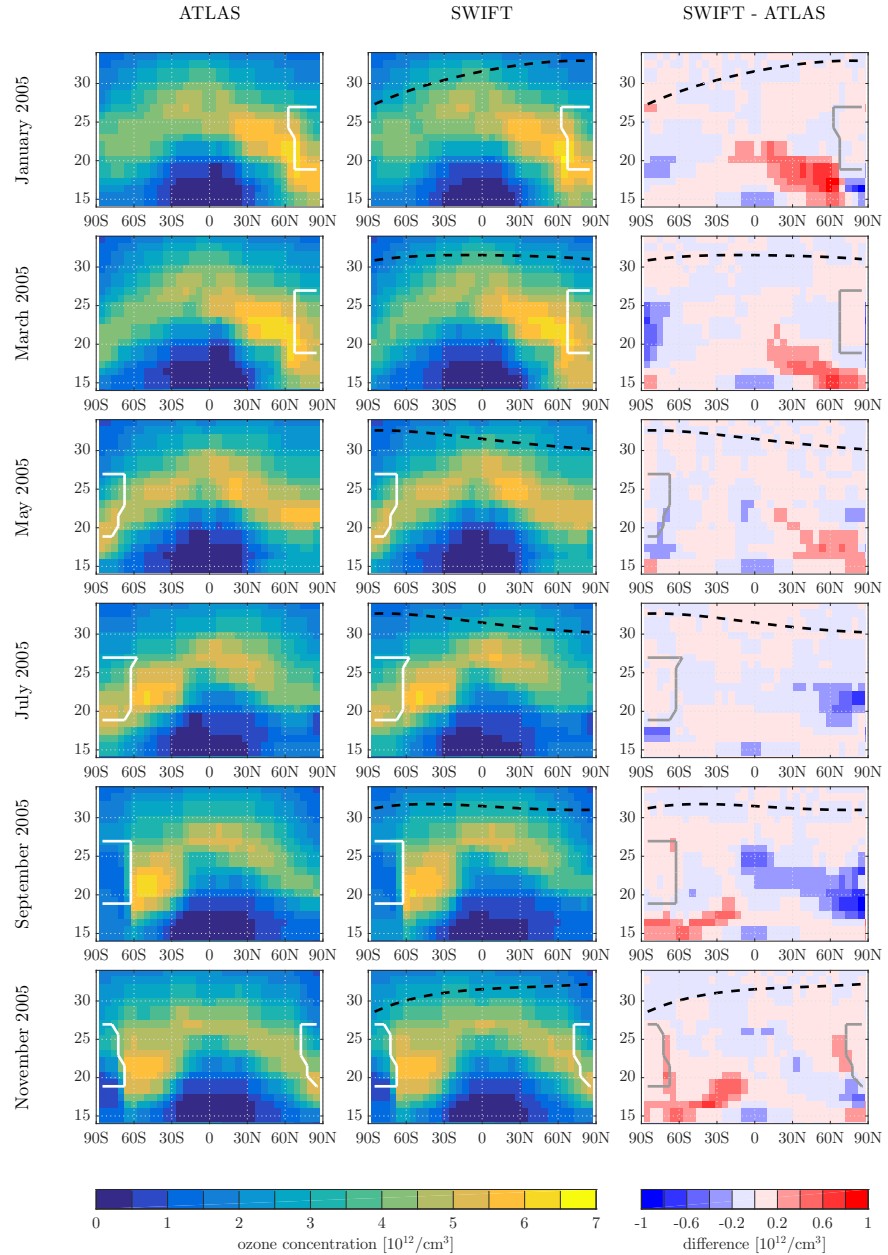

**Figure 5. 2005:** Zonal and monthly mean stratospheric ozone concentrations plotted in equivalent latitude vs. pressure altitude. Left column: reference simulation with ATLAS, middle column: SWIFT simulation, right column: difference. The dashed black contour shows the upper boundary of the $\Delta O_x$-regime. The white contour indicates the location of the polar vortices.

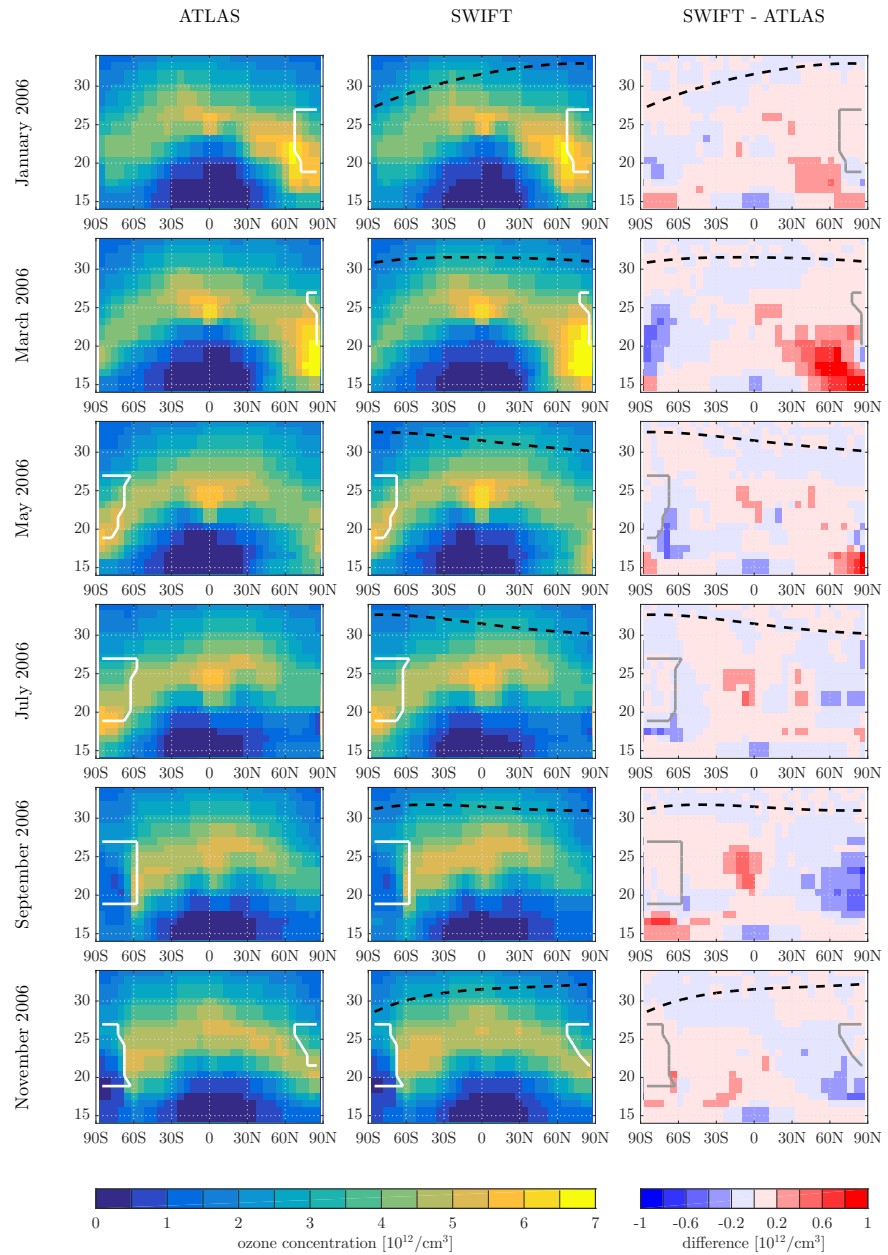

**Figure 6. 2006:** Zonal and monthly mean stratospheric ozone concentrations plotted in equivalent latitude vs. pressure altitude. See also Figure 5.

values below 200 DU). Otherwise, the spread of the dots agrees well in both periods, proving that the SWIFT simulation is not less accurate outside the training data period (pink) than under conditions, which are part of the training data set (green).

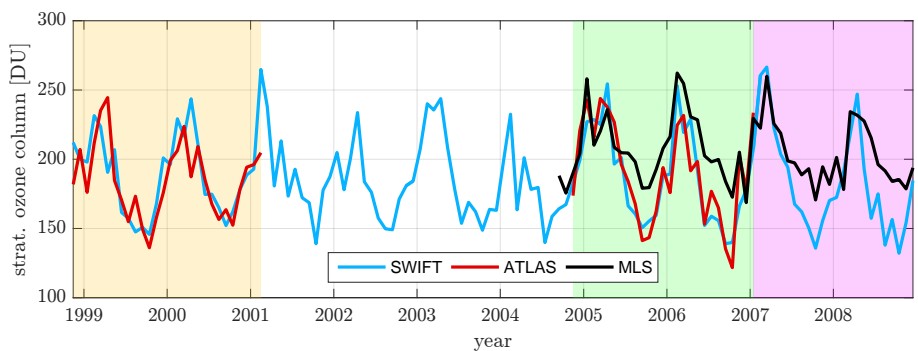

**Figure 7.** Monthly mean values of the stratospheric ozone column (15 km – 32 km) over Potsdam (52.4° North, 13.0° East). The bright blue line shows the continuous 10 year simulation with SWIFT. The orange and green shaded patches are the periods where the training data originates from, hence ATLAS data is available (red line). Beginning in fall 2004, Aura-MLS data is available (black line) and during the pink period SWIFT and MLS are compared outside the training data period.

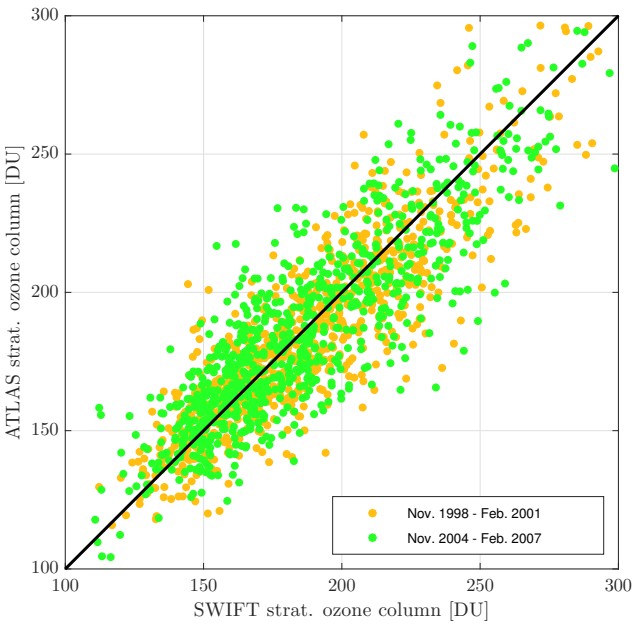

**Figure 8.** SWIFT vs. ATLAS scatter plot of daily averaged stratospheric ozone columns (15 km – 32 km) over Potsdam. The orange and green dots correspond to the two training data periods in Figure 7.

### 5.4 Computational cost of Extrapolar SWIFT

The design of Extrapolar SWIFT enables full parallelization, since individual model nodes can independently evaluate the polynomial functions. A function consists of 30 to 100 polynomial terms, varying from month to month. Per model node and time step 3 polynomial functions have to be evaluated, one domain-polynomial and 2 $\Delta O_x$ polynomial functions for

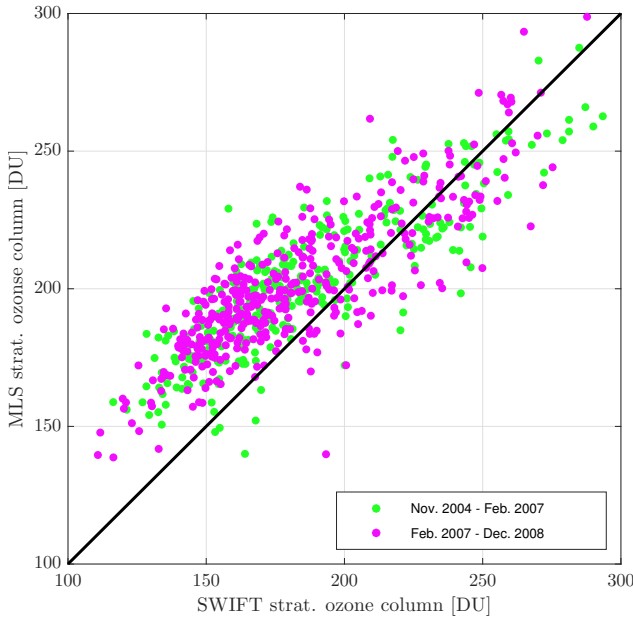

**Figure 9.** SWIFT vs Aura-MLS scatter plot of daily averaged stratospheric ozone columns (15 km – 32 km) over Potsdam. The green dots correspond to the second training data period, and the pink dots correspond to pink period in Figure 7.

the interpolation between 2 months. During the preparation of this paper Extrapolar SWIFT has been coupled to the climate model ECHAM6.3. The Fortran SWIFT code is not fully optimized yet and the current estimates on the computation time are preliminary. An initial estimate of the increase in computation time caused by Extrapolar SWIFT is roughly 10 %. In comparison to an ECHAM version employing full stratospheric chemistry (ECHAM MESSy Atmospheric Chemistry model

5    (EMAC)), the ECHAM + Extrapolar SWIFT requires 6 – 8 times less computation time (only estimated).

The version of Extrapolar SWIFT coupled to the ATLAS CTM is implemented in MATLAB, because the ATLAS model was written in MATLAB. SWIFT in ATLAS is not optimized for speed and the evaluation of the polynomials is computed on a single core. However, when comparing the full stratospheric chemistry scheme of ATLAS versus the evaluation of the SWIFT polynomial functions, the ozone layer can be computed $10^4$-times faster than in the CTM.

10    **6    Conclusions**

The Extrapolar SWIFT model is a numerically efficient ozone chemistry scheme for global climate models. Its primary goal is to enable the interactions between the ozone layer, radiation and climate, while imposing a low computational burden to the GCM it is coupled to. We accomplished this by approximating the rate of change of ozone of the detailed chemistry model ATLAS by algebraic equations. Orthogonal polynomial functions of fourth degree are used to approximate the rate of

15    change of ozone over 24 h. An automated and optimized procedure approximates one globally valid polynomial function to a monthly training data set. In our repro-modelling approach we reduce the dimensionality of the model through exploitation

of covariance between variables. The polynomial functions are a function of only 9 *basic* variables (latitude, pressure altitude, temperature, overhead ozone column, total chlorine, total bromine, nitrogen oxides family, water vapor and the ozone field). At the same time, all physical and chemical processes contained in the full model's output get parameterized in the repro-model.

Running the Extrapolar SWIFT model requires only the 12 monthly polynomial functions and information about the 9
*basic* variables. The domain of the polynomial function is defined by the 9 dimensional training data set. A wide range of stratospheric variability needs to be included in the training data set to increase the robustness of the polynomial functions. We have shown, that the SWIFT model can cope with a certain degree of unknown variability, e.g. induced by climate change. For example, we estimate that the polynomial functions can handle changes of up to 10 % increase or decrease in stratospheric chlorine loading, without adjusting the current training data set. More extreme changes, e.g. a 50 % reduction of chlorine
requires an extension of the training data with values of disturbed chemistry simulations. For handling occasional outliers, i.e. combinations of the 9 *basic* variables outside the domain of definition, Extrapolar SWIFT includes a procedure to prevent extrapolation of the polynomial functions.

Simulations with the Extrapolar SWIFT model coupled to the ATLAS CTM have shown good agreement to the reference model ATLAS. The stability of SWIFT has been proven with a simulation over a 10-year period, in which SWIFT was validated
against model and observational references. Errors did not accumulate over the extended simulation period. Average deviations of the integrated stratospheric ozone column (15 km – 32 km) are $\pm15$ DU between ATLAS and SWIFT. The comparison to Aura MLS measurements showed an equally good agreement with Extrapolar SWIFT, except for the periods of underestimation of the stratospheric ozone column in autumn. This underestimation however is a bias that originates from the source model ATLAS. The computation of the solution of a polynomial function with up to 100 terms is significantly faster than solving a
chemical differential equation system. Extrapolar SWIFT requires $10^4$-times less computation time than the chemistry scheme of the ATLAS CTM.

# 7  Code availability

The source code of the Extrapolar SWIFT model (version 1.0) and the Polar SWIFT model (version 2.0) is available via a publicly accessible Zenodo repository https://zenodo.org/record/1020048. The repository has the DOI: 10.5281/zenodo.1020048.
The ATLAS CTM is available on the AWIForge repository (https://swrepo1.awi.de/). Access to the repository is granted on request. Please contact Ingo.Wohltmann@awi.de. If required, the authors will give support for the implementation of SWIFT and ATLAS.

*Acknowledgements.* This work was supported by the BMBF under the FAST-O3 project in the MiKliP framework programme (FKZ 01LP1137A) and in the MiKliP II programme (FKZ 01LP1517E). This research has received funding from the European Community's
Seventh Framework Programme (FP7/2007–2013) under grant agreement no. 603557 (StratoClim). This study has been supported by the SFB/TR172 "Arctic Amplification: Climate Relevant Atmospheric and Surface Processes, and Feedback Mechanisms (AC)³" funded by the

Deutsche Forschungsgemeinschaft (DFG). We thank ECMWF for providing reanalysis data and the Aura-MLS team for observational data on stratospheric trace gas constituents.

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
