# Peer review of "The Extrapolar SWIFT model (version 1.0): Fast stratospheric ozone chemistry for global climate models"

_Geoscientific Model Development, 2017_

## Short Comment (SC1) · 10 Oct 2017

Daniel

As explained in https://www.geoscientific-model-development.net/about/manuscript_types.htmlGMD is expecting that authors upload the program code of models as a supplement or make the code available at a data repository preferable with an associated DOI (digital object identifier) for the exact model version described in the paper. If for some reason your code cannot make available in this form as the code availability section in your paper suggests you need to state the reasons why the code is not available or access is restricted. Pleas note that in the code accessibility section you

can still point the reader to your web site for updates even if you provide the code as supplement or use a DOI for a release.

There seems to be a vital dependence to the ATLAS CTM code. Can you please clarify which version of ATLAS CTM is required and also how to get access to it?

Many thanks

Lutz Gross GMD Executive Editor

---

## Referee Comment (RC1) · Anonymous Referee #1 · 20 Oct 2017

The Extrapolar SWIFT sub-model is described in this work. The purpose of effort is to develop a stratospheric chemical approach for ozone that is fast, accurate, and consistent with model variability of an underlying GCM. This sub-model can then be used in climate simulations (similar to IPCC type assessments) with a low numerical cost. The authors have done an excellent job of laying out the mathematical and engineering approaches and have quantified the error with multiple comparisons to a full stratospheric chemistry CTM (i.e., ATLAS). The approach is unique and is appropriate for publication in GMD. I highly recommend this work to be published (as is). Below is a comment / suggestion that the authors may choose to address.

The authors show in table 1 that there are 9 "basis variables" that are needed to drive Extrapolar SWIFT (i.e., latitude, pressure, temperature, overhead column ozone, chlorine family, bromine family, nitrogen family, hydrogen family, and odd-oxygen). The ATLAS model is then used to create an "extensive training data set" which is used to create the polynomial functions (based on the 9-dimensional hyperspace) of these basis variables. This seems to work exceptionally well, especially for the comparisons shown in Figure 5 (year 2005), Figure 6 (year 2006), and Figure 7 (years 1999-2009). However, what is interesting about the choice of these years for Extrapolar SWIFT model validation is that these years to not have a major volcanic eruption. It would be useful / interesting (to show) how this sub-model responds over say the 1990-1994 period, which encompasses the Mt. Pinatubo eruptions. Since volcanic eruption and subsequent ozone change are important part of historical record and do have a climatic impact this would (in my opinion) add value to this work.

---

## Author Comment (AC1) · 24 Oct 2017

Dear Lutz Gross,

Thank you for your remarks. Please find our answers and suggestions below.

*"As explained in https:// www.geoscientific-model-development.net/ about/ manuscript_ types.html GMD is expecting that authors upload the program code of models as a sup-*

*plement or make the code available at a data repository preferable with an associated DOI (digital object identifier) for the exact model version described in the paper. If for some reason your code cannot make available in this form as the code availability section in your paper suggests you need to state the reasons why the code is not available or access is restricted."*

The repository (https://swrepo1.awi.de/) specified in the section 7. Code availability of the paper is accessible upon request to the authors. This repository contains the whole ATLAS CTM + SWIFT code. Our initial concern about making the entire ATLAS + SWIFT code publicly accessible was that we do not have the work power to offer support to potential users. However, we decided to make the SWIFT code (polar and extrapolar) available via a publicly accessible **Zenodo repository (https://zenodo.org/record/1020048)**.

This repository contains the version 2.0 of the Polar SWIFT model and the version 1.0 of the Extrapolar SWIFT model:

1. The MATLAB code for the SWIFT integration into the Chemistry and Transport Model (CTM) ATLAS. [`swift_MATLAB.tar.gz`]

2. The Fortran code for the SWIFT integration into a General Circulation Model (GCM), e.g. ECHAM 6. [`swift_fortran.tar.gz`].

3. (only Extrapolar SWIFT) The Polynomial functions stored as NetCDF-files [`swift_extrapolar_polynomials_v1.0.tar.gz`].

4. (only Extrapolar SWIFT) The data processing and fitting routines to determine the polynomial functions. Please refer to the included HTML documentation (./doc/index.html). [`swift_extrapolar_source_v1.0.tar.gz`].

Additionally the Zenodo repository offers a **DOI**: 10.5281/zenodo.1020048

*"There seems to be a vital dependence to the ATLAS CTM code. Can you please clarify which version of ATLAS CTM is required and also how to get access to it?"*

There are several aspects to this question:

1. The training and testing data sets for the polynomial approximation were compiled from ATLAS CTM simulations. After this data collection step the determination of the polynomials is entirely independent of the ATLAS CTM, i.e. the polynomials can be evaluated independently of a specific model frame work. In order to validate and optimize the polynomials we coupled the Extrapolar SWIFT model to the ATLAS CTM, using only its transport and trajectory mixing scheme. But we could have also coupled SWIFT to another model, as we started doing with the ECHAM 6.3 GCM.

2. In order to run the MATLAB SWIFT code (see Zenodo repository) the ATLAS CTM is required, since the routines and functions are interfacing with the ATLAS CTM code. For access and support to the ATLAS CTM code, please contact Ingo.Wohltmann@awi.de. The current ATLAS version in the repository is fully compatible with the SWIFT code. As mentioned before, we can not make the ATLAS code publicly accessible due to the lack of work power for the support of the model.

---

## Referee Comment (RC2) · Anonymous Referee #2 · 30 Oct 2017

Comments on Kreyling et al., The extrapolar SWIFT model...

The manuscript describes the development and testing of a parameterized ozone for use in general circulation models that can be used to provide ozone fields for use in the model radiation that are consistent with the model circulation. The parameterization expresses the total chemical tendency of ozone for the extrapolar mid to lower stratosphere as a function of nine variables that are either derived directly from the model or specified from an external climatology. The paper is very well written, the methodology used to develop the SWIFT parameterization is logically laid out and well described and the approach to reducing both the parameters and the terms in the polynomial

functions appears to be quite rigorous.

Overall, I only have minor comments. Although I do have one nagging question that I cannot quite figure out for myself that perhaps could be addressed more explicitly in the article. As discussed in the manuscript, there are existing approaches to specifying ozone fields that are consistent with the model circulation, such as Linoz or the Cariolle scheme. These schemes rely on linearization of the ozone chemistry around a reference climatology and, for the example of Linoz, only account for variations in temperature, local ozone concentration and overhead ozone column. One of the significant advantages of SWIFT would be the explicit inclusion of variations due to the major chemical families that affect ozone, including HOx, NOy, Cly and Bry. To ensure that the SWIFT parameterization does not extrapolate outside of the range of conditions for which it has been developed, there is a rigorous check on local conditions using an additional polynomial – the domain-polynomial. The problem of a large-scale shift in the conditions for which the SWIFT model has been designed is discussed in the paper and the example of a decrease in Cly of 50% (Page 11, Line 30) is given. Is the SWIFT model really robust to a large, say 30%, change in Cly? I do greatly appreciate that the parameterization has taken advantage of the range of chemical environments within the lower to mid-stratosphere to capture a wide range of conditions, but these different chemical families also have a certain degree of correlation. This does make me wonder how large a variation in Cly would be required before the new combination of HOx, NOy, Cly and Bry falls outside of the range of validity for the SWIFT model. Is there a simple way to test the validity to variations in Cly by using the domain-polynomial and scaling down Cly to find how rapidly parts of the atmosphere begin to fall outside the range of validity? If quantitatively testing this is not as simple as I imagine, feel free to disregard this question though I do sincerely think it would help strengthen the paper.

My other minor comments are given below.

Page 1, Line 9: I might suggest replacing 'local ozone column' with 'overhead ozone

column' as this might be a bit more specific.

Page 2, Lines 6-7: the wording seems to suggest all CMIP5 models used prescribed ozone and I think a more accurate representation would be 'many of the CMIP5 simulations'. A quick read of Eyring et al. (JGR, 118, 5029-5060, doi:10.1002/jgrd.50316, 2013) suggests nine of the 46 models had fully interactive chemistry and a further nine used prescribed ozone calculated from the same GCM.

Page 2, Lines 30-32: Would it be clearer to the reader if this sentence made reference to the 'partitioning' of the chemical families being in photochemical steady state?

Page 5, Lines 1-4: As written I am having trouble following the argument given by: 'In the extrapolar regions the short-lived reactive species (e.g. ClOx or BrOx ) are sufficiently close to chemical equilibrium determined by the local conditions (e.g. pressure, temperature, radiation and the abundance of reaction partners). Consequently, in the chemical families containing only one reservoir gas (NOy and HOy ) the concentration of the short-lived species is uniquely determined by the abundance of the total family.' I agree completely with the statement that the short-lived species are in chemical equilibrium and that the partitioning within the family of short-lived species can be derived from a photochemical steady-state assumption. But I fail to see how this fact can then be used to derive the partitioning between the short-lived and reservoir species for chemical families with only one reservoir gas. For the NOy and HOy families do you not need to first divide the family into the short-lived and reservoir fractions, before chemical equilibrium can be used to further partition the short-lived species? From the way this process is described, it sounds like for the NOy and HOy families you assume local chemical equilibrium between the short-lived and reservoir species and this could be more clearly stated.

Page 17, Line 16: Starting here in Section 5.2 you compare a two year run of the ATLAS CTM using SWIFT where the background states for HOx, NOy, Cly and Bry are taken from daily zonal-average fields from the reference, full chemistry ATLAS run.

Then in Section 5.3 you compare a 10-year simulation of ATLAS-SWIFT using monthly climatologies. Is it then possible to separate the errors that are due to the use of the monthly HOx, NOy, Cly and Bry climatologies by comparing the two-year period that is common to both of these runs? Perhaps just by adding an extra line to Figure 7?

Page 21, Line 13: It is stated here ' An initial estimate of the increase in computation time caused by Extrapolar SWIFT is roughly 10 %.' where I assume that the 10% increase is relative to the ECHAM6.3 using specified ozone - i.e. no chemistry at all? Is there easily available any estimate of the increase in computation time for ECHAM6 when a full stratospheric chemistry is included that could be quoted here?

Page 21, Lines 16-17: I imagine part of the factor of 10ˆ4 difference in speed between the full ATMOS model and the SWIFT ozone is due to the fact that SWIFT model has a significantly reduced number of advected species. It might be worthwhile to mention this as one of the factors in the reduced speed.

---

## Author Comment (AC2) · 8 Nov 2017

Dear Referee 1,

Thank you for your positive feedback and your remark on volcanic activity.

*"It would be useful / interesting (to show) how this sub-model responds over say the 1990-1994 period, which encompasses the Mt. Pinatubo eruptions. Since volcanic eruption and subsequent ozone change are important part of historical record and do have a climatic impact this would (in my opinion) add value to this work."*

As an outlook to Extrapolar SWIFT we wish to include an even wider range of strato-

spheric conditions in the training data sets, e.g. perturbed meridional circulation or a reduction of the chlorine load. Including years with strong volcanic activity should definitely be considered as well.

In the current version of the ATLAS CTM, the background aerosol is based on a fixed climatology based on SAGE II. However, volcanic eruptions are not included in this fixed climatology. The impact of volcanoes on the ozone layer is currently only partly included via the meteorological fields, from e.g. ECMWF ERA-Interim.

However acquiring and incorporating aerosol fields including volcanic activity of the past decades is possible (e.g. CMIP6-Forcings).

---

## Author Comment (AC3) · 16 Nov 2017

Dear Referee #2,

Thank you for your positive review. Please find our answers and adaptations to your remarks below.

1. *"Is the SWIFT model really robust to a large, say 30%, change in Cly?"*

   In future training data set, it is planned to include data from 30% or 50% Cly reduction scenarios, in order to cope with such conditions. Currently, an average reduction of 30% in stratospheric chlorine loading will force the polynomials

   outside the domain of definition in many areas of the 9 dimensional parameter space. 30% reduction is a significant change. It will shift the Cly-PDF (compare to Fig. 2) to far from the training PDF. Considering the current training data set, we assume that the current polynomials can handle a reduction by 10% (however we did not test this).

2. *"I do greatly appreciate that the parameterization has taken advantage of the range of chemical environments within the lower to mid-stratosphere to capture a wide range of conditions, but these different chemical families also have a certain degree of correlation. This does make me wonder how large a variation in Cly would be required before the new combination of HOx, NOy, Cly and Bry falls outside of the range of validity for the SWIFT model. Is there a simple way to test the validity to variations in Cly by using the domain-polynomial and scaling down Cly to find how rapidly parts of the atmosphere begin to fall outside the range of validity? If quantitatively testing this is not as simple as I imagine, feel free to disregard this question though I do sincerely think it would help strengthen the paper."*

   We will try to respond from two different angles:

   (a) In nature the concentrations of Cly, HOy, NOy and Bry are certainly correlated to some degree. However in SWIFT the individual chemical families originate from monthly climatologies. A change to the Cly climatology does not impact the other climatologies. Applying such a change to Cly, without adapting the other chemical families would render unphysical stratospheric conditions. They would most likely lie outside the domain of definition of the model.

   (b) If Extrapolar SWIFT is used in a simulation scenario with (e.g.) 50% reduction of chlorine loading, the training data set must include data from such a scenario (this is not yet the case). This data should originate from a full

stratospheric chemistry model, i.e. the corresponding HOy, NOy and Bry concentrations come together with the reduced Cly concentrations. Thus the domain of definition of the model gets extended in all four dimensions of the chemical families.

3. *"Page 1, Line 9: I might suggest replacing "local ozone column" with "overhead ozone column" as this might be a bit more specific."*

We replaced "local ozone column" with "overhead ozone column".

4. *"Page 2, Lines 6-7: the wording seems to suggest all CMIP5 models used prescribedozone and I think a more accurate representation would be 'many of the CMIP5 simulations'. A quick read of Eyring et al. (JGR, 118, 5029-5060, doi:10.1002/jgrd.50316, 2013) suggests nine of the 46 models had fully interactive chemistry and a further nine used prescribed ozone calculated from the same GCM."*

We included the words "many of" in the sentence. It reads now: "However, a frequently used approach to represent the ozone layer in general circulation models (GCMs) is the use of prescribed zonal mean ozone climatologies, as in many of the Coupled Model Intercomparison Project 5 (CMIP5) simulations ..."

5. *"Page 2, Lines 30-32: Would it be clearer to the reader if this sentence made reference to the 'partitioning' of the chemical families being in photochemical steady state?"*

Yes, mentioning of the keyword "partitioning" might assist the reader in understanding the sentence. But we still want to be specific about the "... diurnal average concentrations of the individual species ...". The sentence now reads: "In extrapolar conditions the diurnal average concentrations of the individual species within the chemical families (partitioning) mentioned . . . "

6. *"Page 5, Lines 1-4: As written I am having trouble following the argument given by: 'In the extrapolar regions the short-lived reactive species (e.g. ClOx or BrOx ) are sufficiently close to chemical equilibrium determined by the local conditions (e.g. pressure, temperature, radiation and the abundance of reaction partners). Consequently, in the chemical families containing only one reservoir gas (NOy and HOy ) the concentration of the short-lived species is uniquely determined by the abundance of the total family.' I agree completely with the statement that the short-lived species are in chemical equilibrium and that the partitioning within the family of short-lived species can be derived from a photochemical steady-state assumption. But I fail to see how this fact can then be used to derive the partitioning between the short-lived and reservoir species for chemical families with only one reservoir gas. For the NOy and HOy families do you not need to first divide the family into the short-lived and reservoir fractions, before chemical equilibrium can be used to further partition the short-lived species? From the way this process is described, it sounds like for the NOy and HOy families you assume local chemical equilibrium between the short-lived and reservoir species and this could be more clearly stated."*

Yes we assume local chemical equilibrium for NOy and HOy between the short-lived and reservoir species. The sentence is appended with: "the concentration of the short-lived species is uniquely determined by the abundance of the total family, i.e. we assume local chemical equilibrium between the short-lived and reservoir species."

7. *"Page 17, Line 16: Starting here in Section 5.2 you compare a two year run of the ATLAS CTM using SWIFT where the background states for HOx, NOy, Cly and Bry are taken from daily zonal-average fields from the reference, full chemistry ATLAS run. Then in Section 5.3 you compare a 10-year simulation of ATLAS-SWIFT using monthly climatologies. Is it then possible to separate the errors that are due to the use of the monthly HOx, NOy, Cly and Bry climatologies by*

*comparing the two-year period that is common to both of these runs? Perhaps just by adding an extra line to Figure 7?"*

The following two plots show the comparison between the stratospheric ozone columns over Potsdam of the 2-year simulation vs. the 10-year simulation. The **Fig. 1** incorporates the 2-yeas simulation (green, 2005 and 2006) in the monthly mean time series (compare to Figure 7 in paper).

The **Fig. 2** shows the same comparison in a scatter plot. X-axis 10-year simulation y-axis 2-year simulation. Instead of monthly means the daily stratospheric ozone columns of the 2005 and 2006 are shown.

Both **Figures** show deviations of 10-20 DU between the 10-year simulation and 2-year simulation. The deviations are comparable to the deviations between ATLAS and the 2-year and ATLAS and the 10-year simulation (see Figure 8 in paper). The use of the daily averaged Cly, Bry, NOy and HOy fields in the 2-year simulation vs. the the actual monthly climatologies in the 10-year simulation does not cause any tendencies in the deviations. Thus it is not possible to attribute higher or lower deviations to either the daily averaged trace gas fields or the monthly climatologies. Since both SWIFT simulations show similar deviations to ATLAS, we do not see additional benefit in adding this information to the paper.

8. *"Page 21, Line 13: It is stated here ' An initial estimate of the increase in computation time caused by Extrapolar SWIFT is roughly 10%.' where I assume that the 10increase is relative to the ECHAM6.3 using specified ozone - i.e. no chemistry at all? Is there easily available any estimate of the increase in computation time for ECHAM6 when a full stratospheric chemistry is included that could be quoted here?"*

We can only give very rough estimates here. The ECHAM model in our working group, which Extrapolar SWIFT was coupled to, does not have an optional full stratospheric chemistry scheme. We consulted our colleagues at FU Berlin

who are running the ECHAM MESSy Atmospheric Chemistry model (EMAC) for some estimates. For one model months EMAC with full stratospheric chemistry (MECCA) requires 60-80 min (114 cores). In the same setup (114 cores) EMAC with prescribed ozone requires 9-10 min per model month. Assumption: EMAC + Extrapolar SWIFT requires 10% more computation time, thus 10-11 min per model months. In conclusion ECHAM + Extrapolar SWIFT is 6 to 8 times faster than ECHAM + full stratospheric chemistry.

9. *"Page 21, Lines 16-17: I imagine part of the factor of 10Ȩ̈4 difference in speed between the full ATMOS model and the SWIFT ozone is due to the fact that SWIFT model has a significantly reduced number of advected species. It might be worthwhile to mention this as one of the factors in the reduced speed."*

The ATLAS CTM is a Lagrangian model. The chemical species are advected along lagrangian trajectories. Thus the computation time for the advection does not scale with the amount of species. The reduced amount of species in SWIFT (coupled to / using the ATLAS advection and mixing scheme) does basically not impact the computation time.

[Figure]

**Fig. 1.** Monthly mean values of the stratospheric ozone column (15 km – 32 km) over Potsdam, Germany. The bright green line shows the 2-year simulation.

[Figure]

**Fig. 2.** Scatter plot showing daily averaged stratospheric ozone columns of the 10-year vs 2-year simulation.

---

## Author Response (AR2)

Dear Fiona

Thank you for your final remarks. Please find our answers and adaptations to your remarks below.

1. *"Reviewer 2, too, was positive about the manuscript although had some comments that required addressing. Firstly, they asked whether the SWIFT model would be robust to a 30% change in Cly loading. Although you answered this question in your response, I felt that you could have included some of your answer in the revised manuscript. In particular, may I ask you to include that the model can cope with upto a 10% change in chlorine loading in your conclusions? It would fit well in the paragraph where you refer to the model being able to cope with some variability and before you discuss extreme values."*

**We added the estimate about the robustness of the polynomials to 10% change in chlorine loading to the conclusion:**

We have shown, that the SWIFT model can cope with a certain degree of unknown variability, e.g. induced by climate change. For example, we estimate that the polynomial functions can handle changes of up to 10% increase or decrease in stratospheric chlorine loading, without adjusting the current training data set. More extreme changes, e.g. a 50% reduction of chlorine requires an extension of the training data with values of disturbed chemistry simulations.

2. *"Secondly, Reviewer 2 asked about the computational cost of the SWIFT model relative to a full chemistry model. Again, although the estimates you have are rough, I would ask that you include something about the relative costs (i.e. 6-8 times faster than ECHAM + full chemistry) in the revised manuscript. A single sentence in Section 5.4 will suffice."*

**We included the comparison of the computation time in the suggested section 5.4.:**

[revised manuscript text omitted]